https://doi.org/10.1038/s41467-019-13955-z · **OPEN**

# Chiral recognition and enantiomer excess determination based on emission wavelength change of AIEgen rotor

Ming Hu[1], Ying-Xue Yuan[1], Weizhou Wang[2], Dong-Mi Li[2], Hong-Chao Zhang[1], Bai-Xing Wu[1], Minghua Liu [3] & Yan-Song Zheng [1]*

Chiral recognition, such as enantioselective interactions of enzyme with chiral agents, is one of the most important issues in the natural world. But artificial chiral receptors are much less efficient than natural ones. For tackling the chiral recognition and enantiomer excess (ee) analysis, up until now all the fluorescent receptors have been developed based on fluorescence intensity changes. Here we report that the chiral recognition of a large number of chiral carboxylic acids, including chiral agrochemicals 2,4-D, is carried out based on fluorescent colour changes rather than intensity changes of AIEgen rotors. Moreover, the fluorescence wavelength of the AIEgen rotor linearly changes with ee of the carboxylic acid, enabling the ee to be accurately measured with average absolute errors (AAE) of less than 2.8%. Theoretical calculation demonstrates that the wavelength change is ascribed to the rotation of the AIEgen rotor upon interaction with different enantiomers.

[1] Key Laboratory of Material Chemistry for Energy Conversion and Storage, Ministry of Education, School of Chemistry and Chemical Engineering, Huazhong University of Science and Technology, Wuhan 430074, China. [2] Beijing National Laboratory for Molecular Science (BNLMS), CAS Key Laboratory of Colloid Interface and Chemical Thermodynamics, Institute of Chemistry, Chinese Academy of Sciences, Beijing 100190, China. [3] College of Chemistry and Chemical Engineering, and Henan Key Laboratory of Function-Oriented Porous Materials, Luoyang Normal University, Luoyang 471934, China. *email: zyansong@hotmail.com

For chiral chemicals in practical usage, such as chiral drugs and chiral agrochemicals, usually one enantiomer of them is effective but other one is invalid or even toxic. Therefore, in order to know the specific physiological activity of each enantiomer, the enantiomeric purity or the enantiomeric excess (ee) should be determined in the process of research, development and manufacture of the chiral chemicals[1–3]. For high throughput discovery of chiral chemicals, many spectroscopic methods including circular dichroism (CD)[4-5], absorption[6,7], fluorescence[7,8], and NMR[9,10] for ee analysis have been reported. Among these methods, fluorescent method based on a chiral fluorescent probe is brought to extremely extensive attention because of its easy accessibility, high cost-effect, high sensitivity and great potential in high throughput assay[7,8,11–30]. For example, Pu et al. have reported a series of BINOL-based fluorescent receptors with excellent performance for ee assay. Some of them showed high selectivity up to several thousand-fold intensity difference between two enantiomers[8,11–17]. Heemstra et al. have used DNA aptamer fluorescent sensor in high throughput ee determination of L-tyrosinamide[18]. James et al. prepared chiral binol-bisboronic acid as fluorescence sensor for ee measurement of a range of sugar acids[19]. By forming chiral covalent organic frameworks (COF)[20], chiral metallacycle[21] and homochiral helicate cage[22], Cui et al. could discriminate and analyze enantiomers of chiral odour vapours such as α-pinene, chiral carboxylic acids and amino acids, respectively. By enantioselective self-assembly, chiral oligophenylenevinylene fluorescent sensor were utilized to measure the ee of 1,2-cyclohexanedicarboxylic acid in Shinkai's group[23]. By using chiral metal-organic framework (MOF) prepared by Lin's group[24] and chiral liquid quantum dots by Li's group[25], the enantiomeric composition of some chiral aminols were analysed. All the reported ee analyses were based on the intensity difference of the fluorescent probe mediated by the two enantiomers, showing high sensitivity up to uM level. However, due to the susceptibility of fluorescence intensity, the accuracy and repeatability of the fluorescent method were usually not high. In addition, it was very rare to give a straight line between fluorescence intensity and ee value[20–25], further leading to measurement errors. Moreover, the ee measurement using these chiral fluorescent receptors was carried out at a given concentration. Consequently, the concentration of the chiral analyte must be known in advance or in rare examples be simultaneously measured using double sensors or a single sensor with multiple-emission bands[11,14,16]. Besides fluorescence intensity changes, if the emission wavelength could also change with external stimuli, the colour difference from the wavelength change will give sharper contrast between the two enantiomers. Moreover, accurate ee measurement will be achieved because of the fewer factors that could affect wavelength change. However, due to challenge of continuous change of emission wavelength with enantiomeric composition, no research work on the ee determination based on emission wavelength change has been reported up to now.

Recently, aggregation-induced emission (AIE) phenomenon has been attracting increasing interest because of its enormous potential in solid emitter and chemo/biosensors[31–34]. By emission intensity change from AIE effect, chiral sensors for chiral recognition and ee determination of many chiral analytes with very high selectivity and very high sensitivity have been developed[26–30,35–39]. According to the well-known AIE mechanism of restriction of intramolecular rotation (RIR)[31], the RIR process not only affects the fluorescence intensity but also tunes the emission colour due to change in conjugation between the central unit (stator) and substituents (rotors) of AIEgens. For example, the most typical mechanochromic effects from AIE molecules are usually ascribed to the different rotation degree of rotors[40–44]. Therefore, this class of chiral AIE rotor has a great anticipation

for accurate determination of enantiomer purity. However, how to utilize the RIR mechanism to design chiral sensors for determining ee values by gradual change of emission wavelength is still a formidable problem.

Here, we report that chiral tetraphenylethylene (TPE) tetramine bearing optically pure amine groups can discriminate between two enantiomers of a number of chiral carboxylic acids including chiral agrochemicals 2,4-D by fluorescent colour difference. Moreover, the accurate determination of ee values of the chiral analytes can be carried out by emission wavelength change and the concentration of the chiral analyte can be also measured by the same probe.

## Results

**Preparation and photophysical properties of chiral probe.** The chiral fluorescent receptors (R)-**6** and (S)-**6** (Fig. 1) were directly synthesized using the commercially available starting material tetrahydroxyTPE. Bearing optically pure amine groups with bulky cyclohexyl substituents, (R)-**6** and (S)-**6** will be able to discriminate between two enantiomers. Meanwhile, due to the large groups at *meta*-position, the repulsion between phenyl rings of the TPE unit is very large and will become even much larger after carboxylic acid is inserted between the phenyl rings due to acid-base interaction. To attenuate the repulsion, the phenyl ring will easily rotate away from the double bond plane and give rise to hypochromic shift of fluorescent colour.

As a pale yellow solid, TPE tetramine (R)-**6** and (S)-**6** emitted green light under 365 nm light (Supplementary Fig. 25). However, they also emitted yellow and green-yellow light in most organic

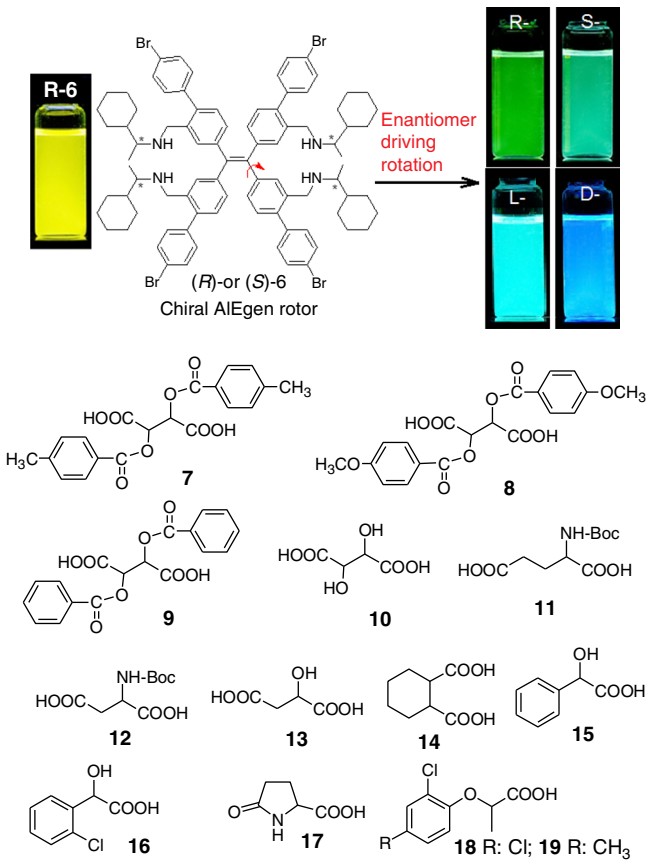

**Fig. 1 Receptor design and enantiomer-driving fluorescent changes.**
Chemical structures of the receptor chiral AIEgen rotors TPE tetramine (R)-**6** and (S)-**6**, and chiral carboxylic acids **7**–**19**.

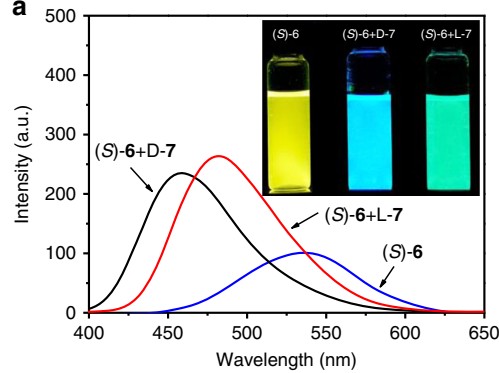

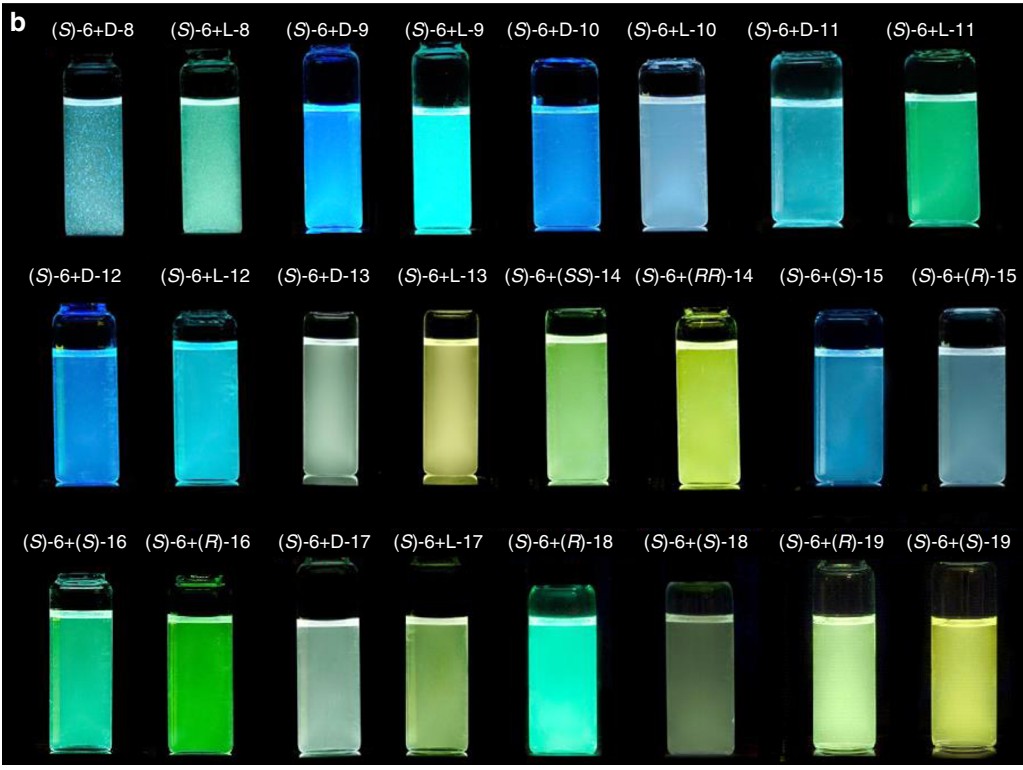

**Fig. 2 Fluorescence changes of receptor with enantiomers. a** Emission spectra of (*S*)-**6** after mixing with two enantiomers of **7** (**7**/(*S*)-**6** 2:1, molar ratio) in cyclohexane/acetone 98:2. Insets and (**b**), photos of solution of (*S*)-**6** without and with chiral acid enantiomers under 365 nm light.

solvents due to the two bulky substituents on each phenyl ring of TPE unit (Supplementary Fig. 26). The tetramines **6** kept the AIE character of TPE unit as revealed by AIE test (Supplementary Fig. 27 and Supplementary Table 1). In addition, as aggregation occurred, the emission displayed a significant hypochromic shift of up to 44 nm from 530 nm to 486 nm. Very interestingly, when hydrogen chloride acid was added into the THF solution of (*S*)-**6**, the fluorescence intensity constantly increased and the emission wavelength was continuously shortened. The hypochromic shift was up to 66 nm from 530 nm to 464 nm, leading to a sharp fluorescent colour change from yellow to blue (Supplementary Fig. 28).

**Chiral recognition of carboxylic acids by (*S*)-6**. Exceptionally, (*R*)-**6** or (*S*)-**6** could discriminate between two enantiomers of a number of chiral carboxylic acids by fluorescent colour difference. After (*S*)-**6** was mixed with di-*p*-toluoyl-D-tartaric acid **7** in a mixed solvent of cyclohexane/acetone 98:2 (volume ratio, the

same below), the emission of (*S*)-**6** was changed from yellow to blue while the mixture of (*S*)-**6** with L-**7** showed green fluorescence (Fig. 2a). The emission maximum wavelength (λ_max) for D-**7** and L-**7** was changed from 536 nm to 457 nm and 483 nm, respectively, showing a difference of 26 nm. Meanwhile, both enantiomers of **7** could enhance the emission intensity of (*S*)-**6** and showed very little intensity difference. Other tartaric acid derivatives including di-*p*-anisoyltartaric acid **8** and dibenzoyl-tartaric acid **9** also aroused blue emission with their D-enantiomers and green emission with L-ones (Fig. 2b and Supplementary Fig. 29). However, the D-enantiomer of tartaric acid **10** gave a brilliant blue while the L-one brought blue grey. For Boc-glutamic acid **11**, Boc-aspartic acid **12**, and malic acid **13**, the D-enantiomer led to blue, sapphire blue and green blue fluorescence while the L-enantiomer gave rise to green, sky blue and pale yellow emission, respectively (Fig. 2b and Supplementary Fig. 29). For 1,2-cyclohexanedicarboxylic acid **14**, it also displayed the colour difference between yellow-green and yellow between by (*S*,*S*)-**14** and (*R*,*R*)-**14**, respectively.

Besides the above dicarboxylic acids, monocarboxylic acids including mandelic acid **15**, 2-chloromandelic acid **16** and pyroglutamic acid **17** also showed different colours between their enantiomers when they were mixed with (*S*)-**6** (Fig. 2b and Supplementary Fig. 29). While (*S*)-**15**, (*S*)-**16** and (*S*)-**17** induced baby blue, green and green grey emissions, (*R*)-**15**, (*R*)-**16** and (*R*)-**17** gave pale iron blue, green-yellow and pale yellow ones, respectively. A difference of 22 nm could be obtained between (*S*)-**16** and (*R*)-**16** while the difference of the decreased fluorescence intensity was small. Very outstandingly, (*S*)-**6** could discriminate between two enantiomers of 2,4-D **18**, one of the most important herbicides (Fig. 2b and Supplementary Fig. 30). (*R*)-**18** shows high herbicidal activity and low toxicity while (*S*)-**18** has no herbicidal activity but high toxicity. Therefore, only enantiopure (*R*)-enantiomer could be used as farm chemicals[45]. When each enantiomer of **18** was mixed with (*S*)-**6** at a 2:1 molar ratio, (*R*)-**18** led to increased fluorescence intensity and hypochromic shift of 40 nm while (*S*)-**18** aroused a decreased fluorescence intensity and hypochromic shift of only 7 nm, displaying a wavelength difference between two enantiomers up to 33 nm. Another herbicide **19** also showed larger hypochromic shift by its (*R*)-isomer than the (*S*)-one.

When D-**7** was gradually added, one new emission band at short wavelength appeared and rapidly increased while the intensity at a long wavelength of 536 nm continuously decreased (Supplementary Fig. 31). Therefore, an isoluminescent point at about 518 nm was observed. Upon addition of 0.3–0.7 equivalents of D-**7**, obvious double emission bands at 459 nm (blue) and at 536 nm (yellow) were observed. Therefore, at 0.5 equivalents of D-**7**, the mixture of (*S*)-**6** with D-**7** gave off a white light. In contrast, the emission spectra of (*S*)-**6** showed hypochromic shift and an intensity increase while no dual emission bands with addition of L-**7** was detected. Meanwhile, the hypochromic shift of the emission band aroused by L-**7** was much smaller than that by D-**7**. The wavelength difference was up to 26 nm between two enantiomers. Even at 8 molar equivalents of **7**, the wavelength difference was still as large as up to 16 nm. But the intensity difference between D-**7** and L-**7** was very small.

**Ee analysis of carboxylic acids by (*R*)-6 and (*S*)-6.** The significant changes of the emission maximum wavelength of **6** with carboxylic acids could be exploited in determining enantiomeric purity of the chiral acids (Fig. 3). By keeping the molar ratio of (*R*)-**6** to the mixture of D-**7** and L-**7** at 1:1, it was found that the emission maximum wavelength of (*R*)-**6** gradually decreased with ee% of L-**7** in a whole range of −100% to 100% (Fig. 3a and Supplementary Fig. 34). The relationship between the wavelength and ee% was a straight line, which could be utilized as a calibration curve for determining the enantiomer content of **7** with an unknown ee%. The average absolute error (AAE) between measured ee values and actual ones was 2.88% ee, which is comparable with the CD method in a high-concentration solution[4,46]. Furthermore, CIE chromaticity diagram disclosed that the CIE coordinate change was linear from −100% ee to 100% ee (Fig. 3d). Using (*S*)-**6** as the receptor, its emission maximum wavelength linearly increased with ee% of L-**7** in a whole range of −100% to 100%, and was mirror symmetrical to the straight line produced by (*R*)-**6**, confirming the chiral recognition. The AAE value brought by (*S*)-**6** was 3.31% ee.

As a monocarboxylic acid and chiral chemical in practical usage, herbicide **18** could also be analysed for its ee by (*R*)-**6** or (*S*)-**6** (Fig. 3b and Supplementary Figs. 35, 36). From −100% ee to 100% ee, the wavelength change with the ee value of (*R*)-**18** and the CIE coordinate change were all linear. The AAE were 2.79% ee and 2.42% ee, respectively, for (*R*)-**6** and (*S*)-**6**, indicating the

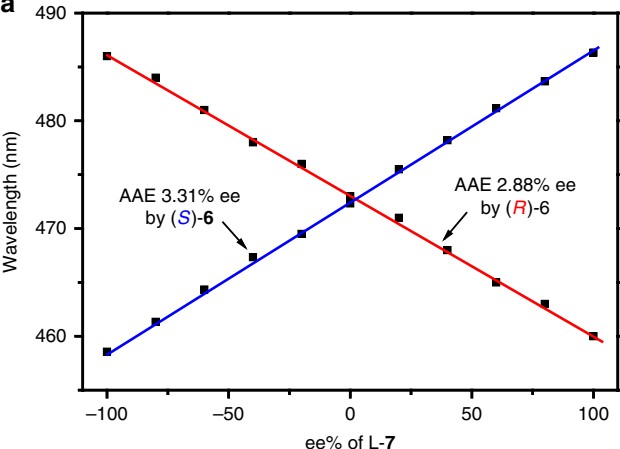

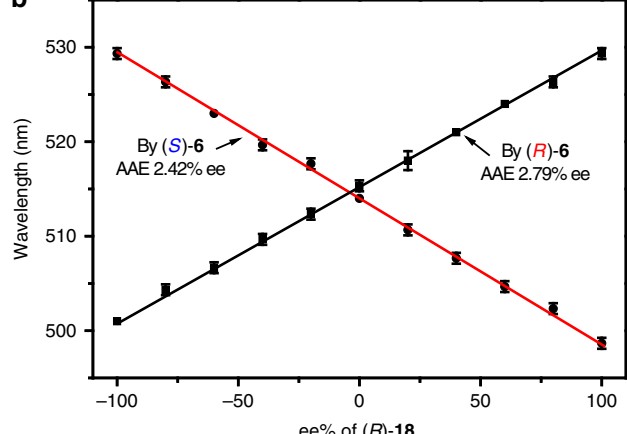

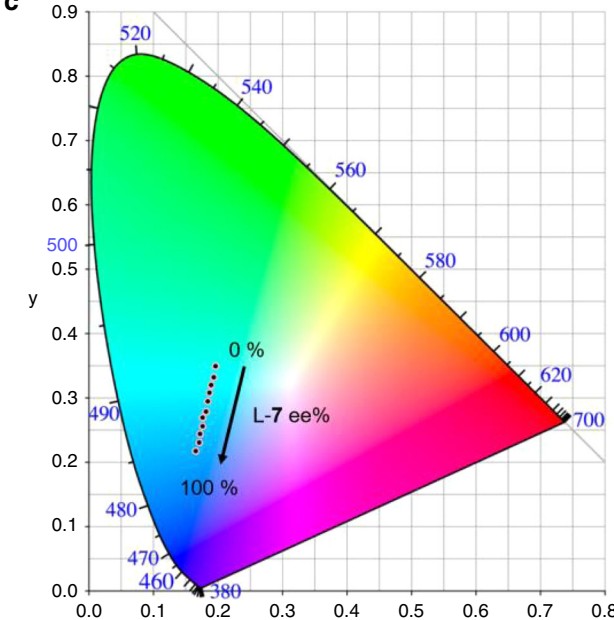

**Fig. 3 Emission wavelength change of receptors with ee.** Change of emission maximum wavelength of **6** with ee% of L-**7** (**a**) and (*R*)-**18** (**b**). **c** CIE chromaticity diagram of ee% of L-**7** measured by (*R*)-**6**. The straight lines were fitted ones by Origin 9.0; Error bars were obtained from three measurements. [(*R*)-**6**] = [(*S*)-**6**] = [**7**] = 1/2[**18**] = 1.0 × 10⁻⁵ M in cyclohexane/acetone 98:2.

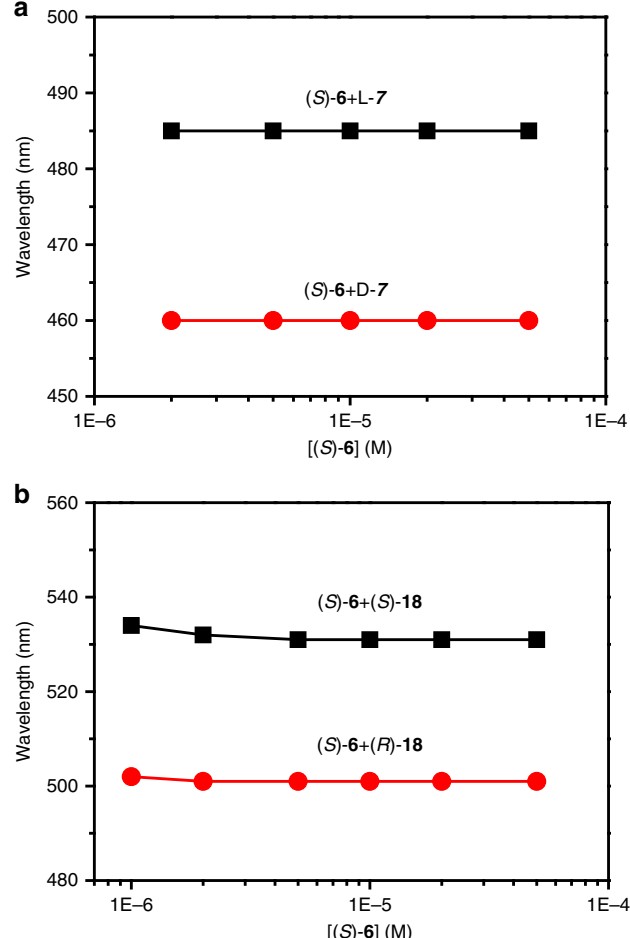

**Fig. 4 Emission wavelength change of receptor with concentration.**
**a** Change of emission wavelength of (S)-**6** with concentration of the mixture of (S)-**6** and one enantiomer of **7** (1:2) and **b** with concentration of the mixture of (S)-**6** and one enantiomer of **18** (1:4) in hexane/acetone 98:2. $\lambda_{ex}$ = 363 nm, em/ex slit widths = 1.5/3 nm.

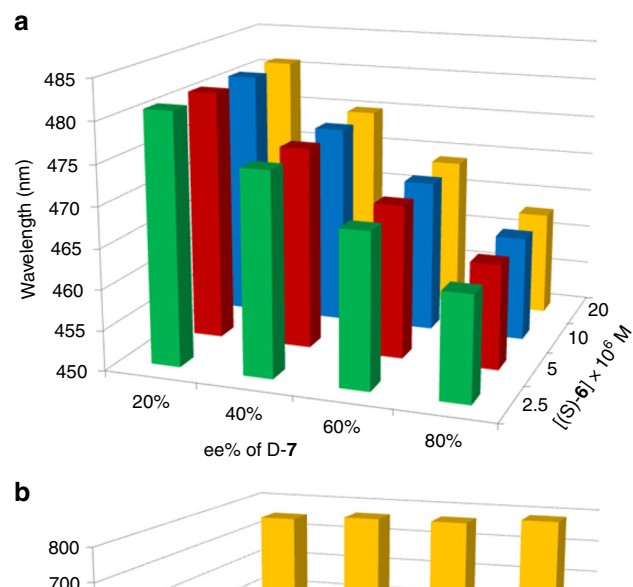

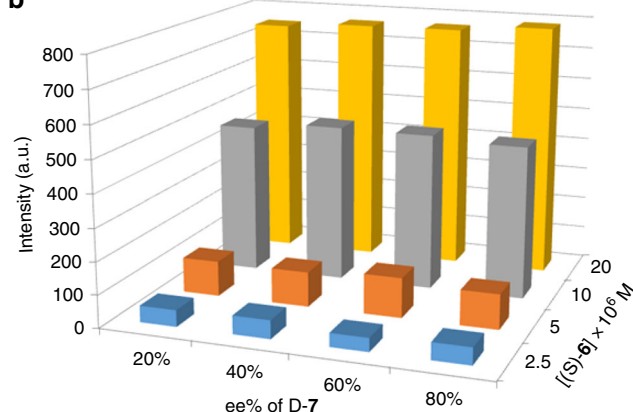

**Fig. 5 Change of wavelength and intensity with ee and concentration.**
**a** Change of emission wavelength and **b** emission intensity of (S)-**6** with enantiomeric composition and concentration of a mixture of (S)-**6** and enantiomers of **7** (1:2) in hexane/acetone 98:2.

high accuracy of the ee analysis. Moreover, the standard deviation for three measurements was less than 1.0, demonstrating that the ee analysis was both accurate and repeatable. Due to the large fluorescence intensity difference of **6** aroused by (R)-**18** and (S)-**18**, the ee analysis based on intensity change was also carried out (Supplementary Fig. 36). The intensity change with ee value of (R)-**18** was approximately linear and mirror symmetrical for the two lines measured by (R)-**6** and (S)-**6**. But the AAE were 13.1% and 8.50% for (R)-**6** and (S)-**6**, respectively, and the standard deviation was about 8.65, which were larger than those from wavelength measurement. In addition, the ee value of 2-chloromandelic acid **16** could also be accurately measured by the emission wavelength change of (R)- and (S)-**6** (Supplementary Fig. 37).

The effect of concentration of a mixture of the fluorescent receptor and chiral analyte on the fluorescence spectrum was studied. As the concentration in cyclohexane/acetone 98:2 increased, D-**7** always aroused much larger hypochromic shift than L-**7** (Fig. 4a and Supplementary Fig. 38). Very unexpectedly, the emission wavelength had almost no change with concentration and the wavelength difference between D-**7** and L-**7** was a constant (25 nm), suggesting that the ee could be determined in a wide range of concentration. In contrast, the emission intensity

increased with concentration but no difference between the two enantiomers was generated. This indicated that the enantiomeric composition of **7** did not affect the fluorescence intensity. Moreover, D-**7**, L-**7** and racemic **7** (D/L-**7**) showed the same intensity change with molar ratio of **7** to the probe or with the concentration of **7** (Supplementary Figs. 31 and 32), demonstrating that the total concentration of **7** could be simultaneously measured through intensity change when ee was determined from wavelength change.

For 2,4-D **18**, as the concentration increased, the wavelength was almost unchanged just like **7**. The wavelength difference between (R)-**18** and (S)-**18** was a constant with a value of 30 nm (Fig. 4b and Supplementary Fig. 39), demonstrating that the ee could be directly determined in a wide range of concentration. But there was an emission intensity difference between the two enantiomers. When the two enantiomers of **16** were tested, the wavelength change with concentration was complicated (Supplementary Fig. 40). From $1.0 \times 10^{-6}$ M to $5.0 \times 10^{-6}$ M and from $3.0 \times 0^{-5}$ M to $1.0 \times 10^{-4}$ M, (R)-**6** gave rise to a larger hypochromic shift than (S)-**6**, but at the middle concentration from $5.0 \times 10^{-6}$ M to $3.0 \times 10^{-5}$ M, (S)-**6** led to a larger hypochromic shift. The intensity difference between the two enantiomers increased with concentration.

As expected, with change of both concentration of (S)-**6** and ee of **7**, the emission wavelength linearly decreased with ee% of D-**7** while no change was observed with concentration of a mixture of (S)-**6** and enantiomers of **7** (Fig. 5a and Supplementary Fig. 41),

**a**

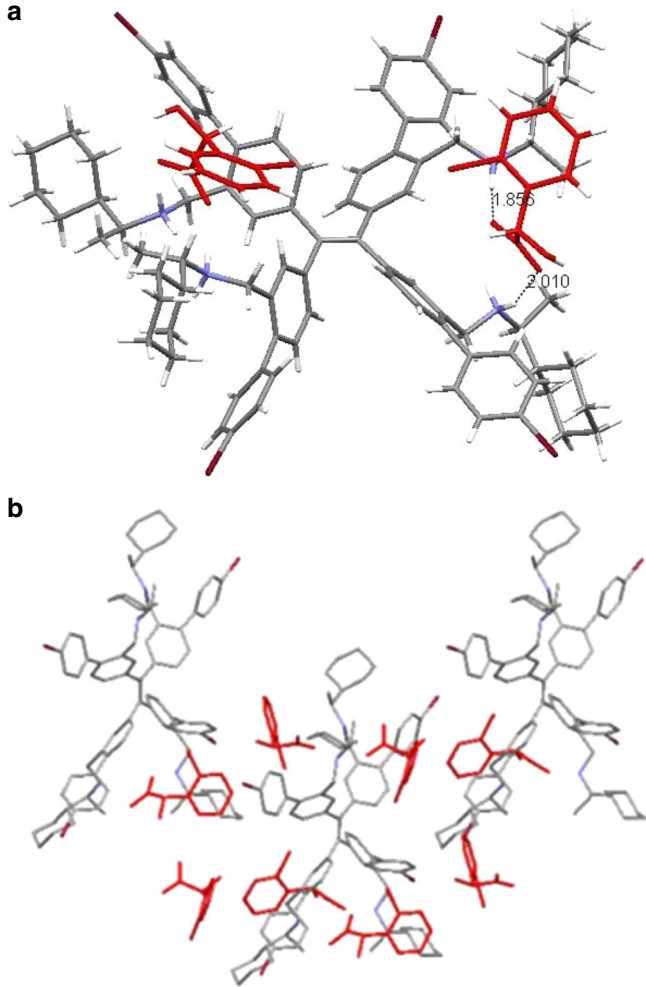

**b**

**Fig. 6 Crystal structure of receptor-enantiomer complex. a** Crystal structure of (*S*)-**6**-(**S**)-**16** complex. **b** The packing mode of molecules of (*S*)-**6**-(*S*)-**16** complex.

confirming that ee could be measured at any concentration. Inversely, the emission intensity was almost constant with ee% of D-**7** but increased with concentration (Fig. 4b and Supplementary Fig. 42). Uniquely, the curve of wavelength change with concentration had a turn point at the same concentration (or the same molar ratio of analyte to probe) for two opposite enantiomers (Supplementary Figs. 31–33 and 35). Consequently, the racemic chiral analyte also had the turn point at the same concentration as that of the enantiomers (Supplementary Fig. 32). Therefore, the total concentration of chiral analyte **7**, **16** and **18** could be obtained from the concentration that corresponded to the turn point on the curve of wavelength change with concentration.

## Discussion

The crystal structure of (*S*)-**6**–(*S*)-**16** complex grown from CHCl₃/acetone/acetonitrile disclosed that the carboxylate anion was inserted between two phenyl rings of TPE unit by simultaneously forming strong hydrogen bonds (1.856 Å and 2.010 Å) with two ammonium groups of (*S*)-**6** (Fig. 6) (The crystal structure data of (*S*)-**6**-(*S*)-**16** and (*S*)-**6**-(*R*)-**16** complex has been deposited in Cambridge Structural Database as CCDC 1921728

and 1954408, respectively). The space between two phenyl rings of the TPE unit was just like a chiral cleft which was more suitable to accept one enantiomer than the other one. Therefore, (*R*)-**6** and (*S*)-**6** displayed an exceptional performance of chiral recognition. Meanwhile, there was a large twist angle of up to 11.90° for the double bond. Moreover, the dihedral angles between the double bond plane and four phenyl rings of TPE unit was large, measured as 52.93°, 52.93°, 46.07°, and 46.07° for the four phenyl rings, respectively, and had an average dihedral angle of 49.46°. These data demonstrated that the repulsive force between phenyl rings of the TPE unit was strong due to insertion of the carboxylic acid between two phenyl rings. To alleviate the steric repulsion, the phenyl rings rotated toward the direction vertical to the double bond, which decreased the conjugation between phenyl ring and double bond, and resulted in hypochromic shift. On the contrary, the crystal of (*S*)-**6**–(*R*)-**16** complex had a space group of P2₁2₁2₁ (Supplementary Fig. 43), which was different from the C121 space group of (*S*)-**6**–(*S*)-**16** complex, demonstrating that different enantiomers resulted in different packing of the same chiral AIEgen. For the (*S*)-**6**–(*R*)-**16** complex, the twist angle of the double bond was 16.29° and dihedral angles of the four phenyl rings was 61.38°, 33.81°, 65.94°, and 42.88°, respectively, with an average value of 51.00°. The crystals of both (*S*)-**6**-(*R*)-**16** complex and the (*S*)-**6**–(*S*)-**16** complex emitted deep blue fluorescence. This result confirmed that the added different enantiomer aroused the different degree of phenyl ring rotation, which should be the main reason why the emission colour changed upon external stimuli.

From the concentration test, the fluorescence intensity of the complex increased with concentration of the complex, confirming the AIE effect of the acid-base complex and the formation of tiny aggregates in the hexane/acetone 98:2 with very low polarity. In addition, the fluorescence intensity of TPE tetramine generally increased upon the addition of the carboxylic acids. Especially with addition of the dicarboxylic acids, the fluorescence was enhanced very obviously because the dicarboxylic acids were easier to arouse aggregation of the acid-base complexes due to larger polarity and probable intermolecular acid-base interactions. Therefore, both the acid-base interactions and the aggregation of the acid-base complexes would cause the rotation of the phenyl rings and colour change just like that in the crystal state.

The absorption maximum wavelength of a mixture of (*S*)-**6** with carboxylic acids also confirmed phenyl ring rotation of the TPE unit (Supplementary Figs. 44 and 45). Compared with (*S*)-**6**, the mixture displayed significant hypochromic shift when (*S*)-**6** was mixed with enantiomers of **7** or **16**. Moreover, D-**7** or (*S*)-**16** aroused larger hypochromic shift than L-**7** or (*R*)-**16** in the mixture, further demonstrating the decrease of conjugation between phenyl rings and the double bond upon addition of acid. To gain a deeper insight into the mechanism of wavelength change, time-dependent density functional theory (TD-DFT) calculations were carried out for a simple model compound tetra(3-methylphenyl)ethylene in the singlet excited state (S₁) at the B3LYP/6-31G(d) level of theory[47,48]. The calculated result disclosed that the energy difference between the first excited state S₁ and ground state increased with the increase of dihedral angle, and the change was even linear (Fig. 7), corroborating that the emission wavelength could be continuously shortened by phenyl ring rotation. In addition, HOMO orbital (−5.37 eV) of (*S*)-**6** located on four nitrogen atoms, but LUMO orbital (−1.61 eV) had no any distribution on the four nitrogen atoms, indicating the probable electron transfer on the nitrogen atoms from ground state to the excited state.

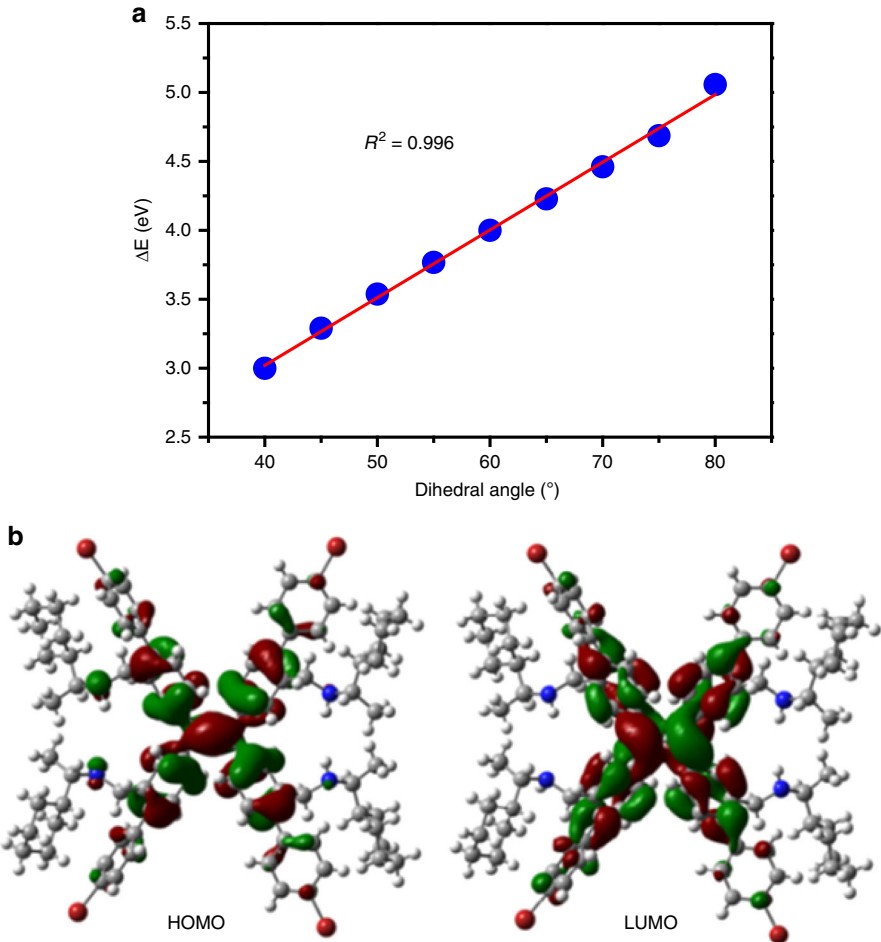

**Fig. 7 Theoretical calculation. a** Change of calculated energy difference ($\Delta E$) between the singlet excited state ($S_1$) and ground state with dihedral angle of model compound tetra(3-methylphenyl)ethylene. **b** HOMO and LUMO orbitals of (*S*)-**6**.

In conclusion, for the first time, two enantiomers of a large number of chiral carboxylic acids were discriminated and the ee determination was carried out by wavelength change instead of fluorescence intensity, showing great potential in accurate high throughput analysis of enantiomeric purity. Exceptionally, the emission wavelength of the chiral fluorescent receptor did not change with concentration of the chiral analytes while it linearly changed with enantiomeric composition. This was very outstanding because the ee value could be accurately determined from the emission wavelength change in a wide range of concentration. Uniquely, besides the total concentration of the analytes could be measured by the turn point on the curve of wavelength changing with concentration, analyte **7** could be simultaneously determined its ee and total concentration by emission wavelength and intensity, respectively, demonstrating the outstanding performance of the chiral AIEgen in chiral analysis. Furthermore, more than 15 kinds of fine fluorescent colours were obtained by chirality tuning of different carboxylic acids, which was unprecedented. Experimental data and theoretical calculation demonstrated that a riot of colour brought by the chirality was ascribed to the continuous rotation of the rotor in the AIEgen. Given that most AIEgens were composed of a stator and a rotor, the constant wavelength change by continuous rotation of the rotor would find more applications in sensors based colour change and tuning of ultra-multiple colours emission.

## Methods

**Materials**. All reagents and solvents were chemical pure (CP) grade or analytical reagent (AR) grade and were bought from China National Pharmaceutical Group Corporation, Aladdin (Shanghai) Bio-Chem Technology Co Ltd, and Meryer (Shanghai) Chemical Technology Co Ltd et al. These reagents and solvents were used as received unless otherwise indicated.

**Measurements**. $^1$H NMR and $^{13}$C NMR spectra were measured on a Bruker AV 400 spectrometer at 298 K in deuterated reagents. Infrared spectra were recorded on Bruker EQUINAX55 spectrometer. Mass spectrum was measured on an Ion Spec 4.7 Tesla FTMS instrument. Absorption spectra were recorded on a Hewlett Packard 8453 UV–Vis spectrophotometer. Fluorescent spectra were collected on a Shimadzu RF-5301 fluorophotometer at 298 K. The single crystal data were collected on Rigaku Saturn diffractometer with CCD area detector. All calculations were performed using the SHELXL97 and crystal structure crystallographic software packages. The fluorescence quantum yield was measured using quinine sulfate ($\Phi_f = 0.546$) in 0.5 M $H_2SO_4$ as standard.

**Preparation of solutions for measurement**. *For fluorescent titration of (S)-6 by hydrochloric acid.* A THF solution of hydrochloric acid was prepared by addition of concentrated hydrochloric acid into THF. Then different equivalents of hydrochloric acids were respectively added into the solution of (*S*)-**6** in THF. The obtained solutions were measured for fluorescent spectra.

*For chiral recognition and ee determination.* A certain amount of (*R*)-**6** or/and (*S*)-**6** and enantiomer of chiral acid was dissolved in acetone, and the resultant solution was diluted with cyclohexane until cyclohexane/acetone 98:2. (*R*)-**6** and (*S*)-**6** are soluble in cyclohexane, but most of chiral acids need to be dissolved in acetone.

**Calculation for molecular orbitals**. All the calculations were performed with the Gaussian 16 package of programs.

## Synthesis of (R)-6 and (S)-6.

*Synthesis of 2.* Trifluoromethanesulfonic anhydride (5.48 mL, 39.37 mmol) was dropwisely added into a stirring solution of pyridine (3.19 mL, 39.37 mmol) and **1**[6] (2.0 g, 3.94 mmol) in dried $CH_2Cl_2$ (50 mL) at 0 °C. After addition was complete, the cooling bath was removed and the mixture was stirred at room temperature for 3 h. The mixture was filtered and evaporated to dryness. The residue was purified by column chromatography (silica gel, ethyl acetate / dichloromethane 1:4) to get pure **2** as white solid (3.67 g, 90%). [1]H NMR (400 MHz, CDCl$_3$) δ 10.14 (s, 4 H), 7.60 (s, 4 H), 7.41–7.23 (m, 8 H). [13]C NMR (101 MHz, CDCl$_3$) δ 185.1, 149.4, 140.8, 137.5, 132.2, 128.7, 123.1, 120.1, 116.9; IR (KBr) *v* 3232, 2922, 1710, 1597, 1430, 1223, 1141, 1085, 887; ESI$^+$ HRMS *m/z* calcd for $C_{34}H_{16}F_{12}O_{16}S_4Na$ 1058.9027 [M + Na], found 1058.8843 [M + Na]$^+$.

*Synthesis of 3.* Tetrakis(triphenylphosphine)palladium (115.6 mg, 0.10 mmol) and k$_2$CO$_3$ (2.01 g, 14.5 mmol) were added into a mixture of **2** (1.5 g, 1.45 mmol) and 4-bromophenylboronic acid (1.74 g, 8.69 mmol) in a mixed solvent of ethyl ether (40 mL) and water (10 mL). The resultant mixture was heated to reflux for 8 h under N$_2$. After reaction was completed, the mixture was filtered and the solid was washed with dichloromethane. The filtrate was dried over anhydrous Na$_2$SO$_4$ before the solvent was evaporated. The resulting residue was purified by column chromatography (silica gel, ethyl acetate/petroleum ether 1:8) to give pure **3** as light

green solid (1.19 g, 58%). Mp 251.9–252.6 °C; [1]H NMR (400 MHz, CDCl$_3$) δ 9.86 (s, 4 H), 7.72 (d, *J* = 1.9 Hz, 4 H), 7.60 (d, *J* = 8.0 Hz, 4 H), 7.37 (dd, *J* = 8.0, 2.0 Hz, 4 H), 7.23 (d, *J* = 8.0 Hz, 12 H); [13]C NMR (100 MHz, CDCl$_3$) δ 191.4, 143.6, 141.8, 140.6, 136.3, 135.9, 133.5, 131.7, 131.5, 130.6, 130.5, 122.9, 77.2; IR (KBr) *v* 3361, 3030, 2841, 2746, 1690, 1474, 1390, 1071, 825 cm$^{-1}$; ESI$^+$ HRMS *m/z* calcd for $C_{54}H_{32}Br_4O_4Na$ 1086.8891 [M + Na], found 1086.8810 [M + Na]$^+$.

*Synthesis of 4.* NaBH$_4$ (733.9 mg, 19.4 mmol) was slowly added into a solution of **3** (1.0 g, 0.97 mmol) in THF/MeOH 4:1 (V/V, 20 mL) under stirring. The resultant mixture was continued to stir at room temperature for 4 h. Then water and dichloromethane were added to give two phases solution. Organic phase was separated and water phase was extracted two times with dichloromethane. The combined organic phase was dried over anhydrous Na$_2$SO$_4$, filtered and evaporated to dryness. The residue was subjected to column chromatography (silica gel, methanol/dichloromethane 1:50) to afford **4** as a white solid (827.8 mg, 80%). Mp 177.7–178.9 °C; [1]H NMR (400 MHz, DMSO-d$_6$) δ 7.60 (d, *J* = 8.0 Hz, 8 H), 7.32 (d, *J* = 7.3 Hz, 12 H), 7.06 (t, *J* = 6.8 Hz, 8 H), 5.13–5.02 (m, 4 H), 4.25 (d, *J* = 5.2 Hz, 8 H); [13]C NMR (100 MHz, DMSO-d$_6$) δ 142.6, 141.0, 139.7, 138.9, 137.6, 131.6, 131.5, 131.3, 129.9, 129.3, 121.0, 61.1; IR (KBr) *v* 3297, 3025, 2925, 2884, 1592, 1477, 1387, 1072, 1004, 821 cm$^{-1}$; ESI$^+$ HRMS *m/z* calcd for $C_{54}H_{40}Br_4O_4$ Na1094.9517 [M + Na], found 1094.9489 [M + Na]$^+$.

*Synthesis of 5.* A mixture of **4** (500 mg, 0.47 mmol) in freshly distilled thionyl chloride (4 mL, 55.6 mmol) was stirred at room temperature for 10 h. After thionyl chloride was evaporated under reduced pressure, water was added. The resultant mixture was extracted with dichloromethane three times. After the combined organic phase was dried over anhydrous Na$_2$SO$_4$, it was filtered and evaporated to dryness. The residue was subjected to column chromatography (silica gel, ethyl acetate / petroleum ether 1:15) to afford **5** as a light green solid (454.3 mg, 85%). Mp 144.2-145.5 °C; [1]H NMR (400 MHz, DMSO-d$_6$) δ 7.66 (d, *J* = 8.0 Hz, 8 H), 7.36–7.27 (m, 12 H), 7.19 (d, *J* = 7.9 Hz, 4 H), 7.12 (d, *J* = 8.1 Hz, 4 H), 4.46 (s, 8 H); [13]C NMR (100 MHz, DMSO-d$_6$) δ 142.4, 140.7, 139.3, 139.0, 134.7, 134.0, 131.8, 131.4, 130.4, 121.5, 44.5; IR (KBr) *v* 3025, 2962, 2867, 1908, 1589, 1477, 1388, 1260, 1072, 1003, 765 cm$^{-1}$; ESI$^+$ HRMS m/z calcd for $C_{54}H_{36}Br_4Cl_4K$ 1182.7901 [M + K], found 1182.7872 [M + K]$^+$.

*Synthesis of (S)-6.* Compound **5** (300 mg, 0.26 mmol), anhydrous K$_2$CO$_3$ (358.8 mg, 2.6 mmol), potassium iodide (431.6 mg, 2.6 mmol), and (S)-(+)-1-cyclohexylethylamine (0.31 mL, 2.08 mmol) were added into a round bottom flask (150 mL) before dry THF (100 mL) was added. The resultant mixture was refluxed for 16 h. After reaction was completed, the mixture was filtered and the solid was washed with dichloromethane. The filtrate was evaporated to dryness and the residue was subjected to column chromatography (aluminium oxide, methanol/dichloromethane 1:50) to afford (S)-6 as a light yellow solid (244.1 mg, 62%). Mp 79.7–82.1 °C; [α]$_D^{25}$ + 20° (c = 5 mg/ml in CHCl$_3$); [1]H NMR (400 MHz, CDCl$_3$) δ 7.51 (d, *J* = 8.1 Hz, 8 H), 7.39 (d, *J* = 8.1 Hz, 8 H), 7.15 (s, 4 H), 7.06 (s, 8 H), 3.51 (d, *J* = 12.2 Hz, 4 H), 3.40 (d, *J* = 12.1 Hz, 4 H), 2.30–2.10 (m, 4 H), 1.80–1.60 (m, 14 H), 1.60–1.40 (m, 8 H), 1.20–1.00 (m, 18 H), 0.70–0.90 (m, 20 H); [13]C NMR (101 MHz, CDCl$_3$) δ 142.9, 140.6, 139.8, 138.8, 137.3, 133.1, 131.1, 131.0, 130.0, 129.4, 121.2, 57.7, 49.4, 42.6, 29.7, 28.0, 26.7, 26.5, 26.4, 16.7; IR (KBr) *v* 3024, 2923, 2850, 1641, 1592, 1477, 1447, 1376, 1102, 1073, 1003, 821 cm$^{-1}$; ESI$^+$ HRMS *m/z* calcd for $C_{86}H_{101}N_4Br_4$ 1509.4719 [M + H], found 1509.4728 [M + 1]$^+$.

*Synthesis of (R)-6.* Compound (R)-6 was synthesized by the same procedure as (S)-6 using (R)-(−)-1-cyclohexylethylamine as reagent. Mp 79.3–81.7 °C; [α]$_D^{24}$ −20.4° (c = 5 mg/mL in CHCl$_3$); [1]H NMR (400 MHz, CDCl) δ 7.51 (d, *J* = 8.0 Hz, 8 H), 7.39 (d, *J* = 8.0 Hz, 8 H), 7.15 (s, 4 H), 7.06 (s, 8 H), 3.51 (d, *J* = 12.2 Hz, 4 H), 3.40 (d, *J* = 12.2 Hz, 4 H), 2.30–2.10 (m, 4 H), 1.80–1.60 (m, 14 H), 1.60 − 1.40 (m, 8 H), 1.20–1.00 (m, 18 H), 0.70 − 0.90 (m, 20 H). [13]C NMR (101 MHz, CDCl$_3$) δ 142.9, 139.8, 138.8, 137.3, 133.1, 131.1, 131.0, 130.0, 129.4, 121.3, 57.7, 49.4, 42.6, 29.7, 28.0, 26.7, 26.5, 26.4, 16.7; IR (KBr) *v* 3024, 2923, 2850, 1641, 1592, 1477, 1447, 1376, 1102, 1073, 1003, 821 cm$^{-1}$; ESI$^+$ HRMS *m/z* calcd for $C_{86}H_{101}N_4Br_4$ 1509.4719 [M + H], found 1509.4730 [M + 1]$^+$.

## Data availability

The data that support the findings of this study are available in Supplemenary Information and from the corresponding author on reasonable request. The crystal structure data of (S)-6-(S)-16 and (S)-6-(R)-16 complex have been deposited in Cambridge Structural Database as CCDC 1921728 and 1954408, respectively.

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

## Acknowledgements

The authors thank National Natural Science Foundation of China (91856125 and 21673089) and HUST Graduate Innovation Fund for financial support, and thank the Analytical and Testing Centre at Huazhong University of Science and Technology for measurement.

## Author contributions

Y.-S. Z. conceived the idea. M. H., Y.-X. Y., D.-M. Li, H.-C. Z., and B.-X. W. performed the experiments. W. W. made the theoretical calculation. Y.-S. Z., M. L., and M. H. wrote the paper.

## Competing interests

The authors declare no competing interests.
