## [Peer Review File · Nature Communications]

Reviewers' comments:

Reviewer #1 (Remarks to the Author):

Chiral recognition and rapid enantiomer excess (ee) analysis are still a challenge in research and development of chiral drug, chiral agrochemicals and other chiral materials. Due to high sensitivity and quickness, a large number of chiral fluorescent receptors for chiral recognition and ee determination have been reported. All these fluorescent receptors are based on fluorescence intensity change rather than emission wavelength change. Although few examples have used fluorescence color change to discriminate between enantiomers of limited chiral analytes, the ee assay from fluorescent wavelength change is not reported up to now. This paper ingeniously exploits intramolecular rotation of AIEgen rotor that can change emission wavelength to fulfill chiral recognition and the ee analysis. The novel chiral AIEgens not only discriminate between enantiomers of a large number of chiral acids, but also can furnish accurate ee analysis. Moreover, the ee can be determined without the need of measuring the total concentration in advance. This research work makes a significant breakthrough in fluorescent ee analysis because it overcomes the problems of low accuracy and extra concentration measurement. It also demonstrates the unique advantage of AIEgens as chemo/biological sensors. Therefore, I strongly recommend this paper to be published in this journal after minor revisions.

1. It is noted that there are four bromine atoms in chiral fluorescent receptors (R)-6 and (S)-6. What is the role of them in the chiral recognition?
2. The chiral recognition and ee analysis can be carried out in highly diluted solution. In this diluted solution, the mixture of the chiral fluorescent receptor and the enantiomer of chiral analyte produces aggregates or is still highly dispersive?
3. It is very interesting that the emission wavelength is not changed with concentration but only linearly changed with ee value. To better demonstrate the emission wavelength change independent of concentration, Fig. S37D and Fig. S38C should be moved to the main text from SI.
4. In Fig. 3, there is a redundant sentence with a spelling error "The straight lines was fitted ones by Origin 9.0". In line 194, the "analyze" should be "analyte".

Reviewer #2 (Remarks to the Author):

This paper reports the use of a tetraphenylethene-based chiral tetraamine (R- or S-6) for the fluorescent recognition of chiral carboxylic acids. The authors found that this probe exhibits different emission wavelengths when treated with the two enantiomers of a chiral acid. Thus, the change in emission wavelength can be used to determine the ee of various chiral acids. This is a new and very interesting discovery. It provides a new class of fluorescent probes for enantioselective recognition of chiral compounds. However, major revisions are needed before this paper can be published as described below.

The authors claim that the probe can be used to determine the ee of a chiral acid without knowing the concentration of the acid because the emission wavelength does not change with concentration. Although in Figure S37 the authors show that the emission wavelength does not change with concentration, these measurements were conducted by maintaining the ratio of the probe versus the analyte at 1:1. Therefore, it is necessary to know the concentration of the analyte in order to identify the proper ratio of the probe versus the analyte for the ee measurement. Figure S37 cannot be used to determine the concentration of the analyte since they were obtained by fixing the ratio of S-6 versus 7 at 1:1 while changing the overall concentration. The authors should add a concentration determination method in this paper.

Figure S31 shows that when the ratio of L- or D-7 versus S-6 was less than 0.5, the two enantiomers of the analyte generated similar changes in the emission wavelength which cannot be used to determine the ee. Only at the higher ratio of L- and D-7 versus S-6 can the change of the emission wavelength be used to determine the ee. This demonstrates that one needs to know the concentration of the analyte in order to choose the proper ratio of the probe versus the analyte before the ee can be determined.

The synthesis of compound 1 or its commercial source should be provided.

Reviewer #3 (Remarks to the Author):

In this manuscript, Zheng et al report the chiral recognition and sensing of enantiomers of different carboxylic acids based on fluorescent color change of TPE derivatives. The fluorescence wavelength of the acceptor show a linear changed with ee of the acid, enabling determination of the ee values. Single crystal structures of two host-guest adducts have been determined by single-crystal X-ray diffraction. DFT calculations suggest that the fluorescent color change was ascribed to the continuous rotation of the rotor in the AIEgen rotor. The result is interesting and can be published in NC after addressing the following issues.

- 1.The author used AIEgens, but the result is not related to AIE. Please just use TPE an delete AIEgen.
- 2.The life times of the acceptor and related host-guest adducts should be provided.
- 3.More discussions should be given to support the mechanism of the fluorescent color change (the solid-state structure is different from the solution structure).

A response to the reviewers' comments point by point.

For Reviewer #1:

1. It is noted that there are four bromine atoms in chiral fluorescent receptors (R)-6 and (S)-6. What is the role of them in the chiral recognition?

The bromine atoms are introduced to link with other aromatic units through Suzuki reaction for further increasing steric volume of the substituents. But it is found that the TPE derivative at present state shows the excellent performance in chiral recognition, so that no further modification is done and the bromine atoms are retained. They should have no special role in chiral recognition.

2. The chiral recognition and ee analysis can be carried out in highly diluted solution. In this diluted solution, the mixture of the chiral fluorescent receptor and the enantiomer of chiral analyte produces aggregates or is still highly dispersive?

It is not sure if the interaction of TPE tetramine (R)-6 and (S)-6 with chiral carboxylic acid produces aggregates because no nanoparticles are observed by dynamic light scattering experiment in the highly diluted solution. However, from the concentration test, the fluorescent intensity of TPE tetramine is increased with the addition of the carboxylic acids especially dicarboxylic acids or with the increase of the concentration. Therefore, the aggregates should be formed after the acid-base interactions.

3. It is very interesting that the emission wavelength is not changed with concentration but only linearly changed with ee value. To better demonstrate the emission wavelength change independent of concentration, Fig. S37D and Fig. S38C should be moved to the main text from SI.

The Fig. S37D and Fig. S38C have been moved into the main text.

4. In Fig. 3, there is a redundant sentence with a spelling error "The straight lines was fitted ones by Origin 9.0". In line 194, the "analyze" should be "analyte".

The typos have been corrected.

For Reviewer #2:

1. The authors claim that the probe can be used to determine the ee of a chiral acid without knowing the concentration of the acid because the emission wavelength does not change with concentration. Although in Figure S37 the authors show that the emission wavelength does not change with concentration, these measurements were conducted by maintaining the ratio of the probe versus the analyte at 1:1. Therefore, it is necessary to know the concentration of the analyte in order to identify the proper ratio of the probe versus the analyte for the ee measurement. Figure S37 cannot be used to determine the concentration of the analyte since

they were obtained by fixing the ratio of S-6 versus 7 at 1:1 while changing the overall concentration. The authors should add a concentration determination method in this paper.

Firstly, the total concentration of D-7 and L-7 can be measured by intensity change with concentration shown in Supplementary Fig. 31 because both D-7 and L-7 show the same intensity change with concentration. To further demonstrate this, new titration curves of intensity change with concentration of D-7, L-7 and even racemic 7 (D/L-7) are measured (Supplementary Fig. 32). It is true that the D/L-7 also shows the same intensity change as D-7 and L-7. Therefore, the intensity change with concentration can be used to measure the total concentration.

Secondly, the total concentration of analyte can be measured by turn point of wavelength change with concentration. It is found that the curve of wavelength change with concentration has a turn point at the same concentration (or same molar ratio of analyte to probe) for two opposite enantiomers as shown in Supplementary Fig. 31, Fig. 32, Fig. 33, and Fig. 35. For example, in Supplementary Fig. 32, D-7 and L-7 have the turn point at the same concentration of about 0.75×10^{-5} M, arousing the turn point for D/L-7 at the same concentration of about 0.75×10^{-5} M. For example, D-7 and L-7 have the turn point at the same concentration of about 0.75×10^{-5} M (Supplementary Fig. 32), arousing the turn point for racemic 7 (D/L-7) at the same concentration of about 0.75×10^{-5} M. Using chiral 7 whose concentration is unknown but should have the much larger concentration than 10^{-5} M level of the probe as sample, it can be diluted to a series of solutions having different concentration. Upon addition of the same amount of probe, the curve of wavelength change with the concentration is measured and the concentration of 7 can be determined by one concentration that corresponds to the turn point.

2. Figure S31 shows that when the ratio of L- or D-7 versus S-6 was less than 0.5, the two enantiomers of the analyte generated similar changes in the emission wavelength which cannot be used to determine the ee. Only at the higher ratio of L- and D-7 versus S-6 can the change of the emission wavelength be used to determine the ee. This demonstrates that one needs to know the concentration of the analyte in order to choose the proper ratio of the probe versus the analyte before the ee can be determined.

The concentration of the chiral analyte can be determined by the methods in the above response to the comment 1. As pointed out by this reviewer, the concentration of the analyte needs to be determined in order to choose the proper ratio of the probe. Therefore, the description about the ee can be determined without known concentration in advance is removed.

3. The synthesis of compound 1 or its commercial source should be provided.

The method for the synthesis of compound 1 has been cited in SI.

For Reviewer #3:

1.The author used AIEgens, but the result is not related to AIE. Please just use TPE an delete AIEgen.

The fluorescent probe **6** is a typical AIEgen because its fluorescence is enhanced very much when water is added into the solution of it in THF (Supplementary Fig. 27). From the concentration test, the fluorescent intensity of TPE tetramine is increased with the addition of the carboxylic acids, especially with addition of the dicarboxylic acids or with the increase of the concentration. This indicates that the aggregates should be formed after the acid-base interactions. It can be explained by the larger polarity of the acid-base complex than the acid monomer or the base monomer so that the complex will have smaller solubility and arouse aggregation in nonpolar solvent hexane:acetone 98:2. The fluorescence enhancement with concentration of the complex should be ascribed to the increased aggregation.

2.The life times of the acceptor and related host-guest adducts should be provided.

The fluorescent lifetimes of the acceptor and related host-guest adducts have been provided.

3.More discussions should be given to support the mechanism of the fluorescent color change (the solid-state structure is different from the solution structure).

More discussions have been added in the paper. In the solvent hexane:acetone 98:2 that has very small polarity, the acid-base salt complex with a large polarity should have very small solubility. Although large aggregates are not observed, the very tiny aggregates of the acid-base complexes that can disperse well in the solvent should exist in the highly diluted solution. From the concentration test, the fluorescent intensity of TPE tetramine is increased with the addition of the carboxylic acids especially the dicarboxylic acids or the fluorescent intensity of the complex is increased with concentration of the complex, indicating AIE effect of the acid-base complex and formation of tiny aggregates in hexane/acetone 98:2. Both the acid-base interactions and the aggregation of the acid-base complexes will give rise to the rotation of the phenyl rings and color change just like that in the crystal state.

REVIEWERS' COMMENTS:

Reviewer #1 (Remarks to the Author):

The authors have carefully revised their manuscript and adequately addressed my previous comments and suggestions. The revisions are satisfactory and the changes are acceptable. The quality of the manuscript has been improved after revision. I do not have further criticism of the work.

Reviewer #2 (Remarks to the Author):

In the conclusion section of this revised version, the authors added the following method to determine the concentration of amino acids: "the total concentration of the analyte can be measured by the turn point on the curve of wavelength change with concentration." With this method, for each unknown acid sample, they need to determine the total concentration of the acid by a series of dilution and titration to find the turning point in the wavelength change in order to find the proper ratio of the acid versus the probe before the ee measurement. In a high throughput analysis, there are a large number of samples need to be measured. It is not possible to carry out a series of dilution and titration for each sample in order to determine the concentration. The problem with this probe is that they have to determine the concentration of the analyte sample, then choose the proper probe concentration before they can measure the ee. That is, for each different sample, they need to use a different probe concentration. This is not practical at all. The authors need to address this problem before publication of this paper.

Following are additional suggested revisions:

1. The authors state that "due to susceptibility of fluorescent intensity, the accuracy and repeatability of the fluorescent method is not high" should be removed since many fluorescence intensity-based methods can give accurate and reproducible results.
2. In literature, there are many reported probes that show both intensity and emission wavelength differences when treated with the enantiomers of a chiral analyte. Therefore, the statement that "for the first time the chiral recognition ... carried out based on fluorescent color change" is not correct and should be removed.

A response to the reviewers' comments point by point.

For Reviewer #1:

The authors have carefully revised their manuscript and adequately addressed my previous comments and suggestions. The revisions are satisfactory and the changes are acceptable. The quality of the manuscript has been improved after revision. I do not have further criticism of the work.

No further concern needs to be addressed.

For Reviewer #2:

In the conclusion section of this revised version, the authors added the following method to determine the concentration of amino acids: "the total concentration of the analyte can be measured by the turn point on the curve of wavelength change with concentration." With this method, for each unknown acid sample, they need to determine the total concentration of the acid by a series of dilution and titration to find the turning point in the wavelength change in order to find the proper ratio of the acid versus the probe before the ee measurement. In a high throughput analysis, there are a large number of samples need to be measured. It is not possible to carry out a series of dilution and titration for each sample in order to determine the concentration. The problem with this probe is that they have to determine the concentration of the analyte sample, then choose the proper probe concentration before they can measure the ee. That is, for each different sample, they need to use a different probe concentration. This is not practical at all. The authors need to address this problem before publication of this paper.

In drug discovery, one lead compound can not be modified too much, otherwise it will lose its activity. In catalyst scanning for asymmetric synthesis, one catalyst is often efficient on one class of substrate but is inapplicable to other class of substrates. Therefore, although there are a large number of samples that need to be measured in a high throughput analysis, these samples are usually the same class of compounds only with different substituent. In this regard, if the probe/analyte ratio is suitable for one molecule, it will be suitable for other molecules in a series of the same class of chiral analytes. Consequently, we do not need to carry out a series of dilution and titration for each sample in order to determine the concentration and then obtain the proper probe/analyte ratio in the measurement of enantiomeric composition, especially in the case that our probe has a wide range of probe/analyte ratio that can be chosen for ee

determination (see supplementary Fig. 32, 33 and 35).

In fact, the measurement of the concentration is much easier than that of the enantiomer excess. For example, by measuring the absorption spectrum, the concentration of the chiral analytes can be facilely determined.

Following are additional suggested revisions:

1. The authors state that “due to susceptibility of fluorescent intensity, the accuracy and repeatability of the fluorescent method is not high” should be removed since many fluorescence intensity-based methods can give accurate and reproducible results.

The sentence “due to susceptibility of fluorescent intensity, the accuracy and repeatability of the fluorescent method is not high” is changed into “due to susceptibility of fluorescent intensity, the accuracy and repeatability of the fluorescent method is usually not high”.

2. In literature, there are many reported probes that show both intensity and emission wavelength differences when treated with the enantiomers of a chiral analyte. Therefore, the statement that “for the first time the chiral recognition ... carried out based on fluorescent color change” is not correct and should be removed.

The phrase “for the first time” has been removed. Nevertheless, the ee measurement based on emission wavelength change is truly not reported up to now.